# GENERATIVE MODELING OF BATTERY DEGRADATION WITH FLOW MATCHING AND DIFFUSION TRANSFORMERS

## ABSTRACT

Battery degradation remains a critical challenge in the pursuit of green technologies and sustainable energy solutions. Despite significant research efforts, predicting battery cycle life accurately remains difficult due to the complex interplay of aging and cycling behaviors. To address this challenge, we introduce FlowBatt, a general-purpose model for battery degradation prediction and synthesis trained via flow matching. FlowBatt leverages a scalable diffusion transformer (DiT) backbone, enabling high expressivity and scalability. The model operates as a probabilistic predictor of entire cycle life trajectories and as a generative model capable of synthesizing realistic degradation curves for data augmentation. We demonstrate the advantages of flow-based generative approaches by comparing models trained with flow matching, diffusion processes, and supervised learning. FlowBatt achieves results that are comparable to or better than state-of-the-art performance for the remaining useful life prediction task and provides accurate and generalizable state-of-health predictions while capturing uncertainty in aging dynamics. Beyond prediction accuracy, this work advances the development of foundational and scalable models for battery degradation.

## 1 INTRODUCTION

### 1.1 LITHIUM-ION BATTERIES

Lithium-ion (Li-ion) batteries are key technologies in the field of energy storage, with applications spanning portable electronics and electric vehicles (Blomgren, 2017). The prominence of these batteries is largely attributable to their high energy density, which enables substantial energy storage within a compact and lightweight form factor. Moreover, Li-ion batteries demonstrate an extended cycle life compared to other battery technologies, quantified in terms of charge and discharge cycles, thereby enhancing their cost-effectiveness for long-term usage. The low self-discharge rate of Li-ion batteries ensures lower energy loss during periods of inactivity than other battery technologies, which is a significant advantage (Galeotti et al., 2015; Vetter et al., 2005). Nevertheless, several challenges remain. Safety continues to be a major issue, as mechanical damage or improper handling can potentially lead to hazardous events such as thermal runaway (Chombo and Laoonual, 2020). Furthermore, the economic and environmental implications of Li-ion battery production, recycling, and disposal present additional complexities that warrant ongoing investigation (Blömeke et al., 2022; Ginster et al., 2024). One persistent challenge that continues to impact their long-term performance and reliability is capacity degradation.

### 1.2 CAPACITY DEGRADATION

The phenomenon of capacity degradation in Li-ion batteries is a multifaceted issue that encompasses both the effects of aging and the effects of cycling. Aging behavior, which is often referred to as calendar aging, pertains to the decline in battery performance over time, irrespective of active usage. Factors such as ambient temperature, state of charge, and storage conditions play a significant role in this degradation mode. In contrast, cycling behavior, also termed cycle aging, is linked to the deterioration that batteries experience during charge and discharge cycles. High charge-discharge

rates and frequent cycling result in the accumulation of irreversible changes within the battery's electrochemical structure. This degradation is driven by a number of factors, including the formation of a solid electrolyte interphase layer, electrolyte decomposition, and the growth of lithium plating (Broussely et al., 2005; Edge et al., 2021). Both aging and cycling behaviors collectively result in overall degradation, reducing the battery's capability to store and deliver electric charge (Rubenbauer and Henninger, 2017) over its operational lifespan. In addition, the aging of batteries is very individual depending on, e.g., usage behavior and environmental conditions, which makes a basic understanding difficult and hinders individual battery management. Despite extensive research, accurately predicting the rate and extent of capacity loss remains a formidable challenge (O'Kane et al., 2022). Therefore, advanced modeling techniques, including machine learning, are increasingly being employed to provide more accurate predictions of the battery degradation.

## 1.3 MACHINE LEARNING IN BATTERY LIFE PREDICTION

The degradation of a battery can be quantified by using key performance indicators, including the state of health (SOH) and the remaining useful life (RUL) (Li et al., 2022). The SOH is a measure of the current capacity of a battery relative to its original capacity, expressed as a percentage, and provides insight into the extent of capacity degradation (Cui et al., 2022; Ren and Du, 2023). In contrast, the RUL is a predictive measure that estimates the remaining operational cycles of a battery before it reaches a defined performance criteria (Li et al., 2023). Methods for battery degradation modeling can be classified according to Rauf et al. (2022) into four domains: i) physics-based models, ii) empirical models, iii) data-driven methods (DDMs), and iv) hybrid methods. Among these various domains, DDMs are emerging as a prominent technique for developing battery degradation models. This is due to the flexibility and independence from specific model assumptions that these approaches offer. In the domain of DDMs, machine learning (ML) methods are widely regarded as one of the most effective approaches for estimating RUL and SOH, due to their ability to address non-linear problems (Rauf et al., 2022). Since all battery RUL and SOH prediction tasks are effectively regression problems, supervised learning is the most commonly used approach in ML battery studies.

Recent literature reviews (Li et al., 2022; 2023; Rauf et al., 2022; Ren and Du, 2023; Wang et al., 2021) indicate that various ML methods are utilized for modeling battery degradation. In the area of artificial neural networks, shallow neural networks can capture nonlinear relationships among an arbitrary number of inputs and outputs, however, they are hindered by slow training processes and a propensity to converge at local minima (Li et al., 2022; Ren and Du, 2023). In contrast, deep learning algorithms demonstrate superior performance in managing large datasets due to their specialized architectures. They provide higher accuracy and enhanced generalization capabilities but incur significant computational costs (Ren and Du, 2023). Techniques such as convolutional neural networks (CNNs), recurrent neural networks, and long short-term memory (LSTM) networks are commonly employed in this context (Li et al., 2023; Rauf et al., 2022; Wang et al., 2021). Additionally, support vector machines achieve a commendable balance between generalization capability and prediction accuracy. However, they may struggle with scalability on larger datasets (Ren and Du, 2023). Similarly, relevance vector machines have the disadvantage of requiring extensive datasets, which results in significant computational complexity. However, they offer the advantage of high accuracy, robust learning capabilities, and the capacity to generate predictions with associated probability distributions (Rauf et al., 2022). Lastly, Gaussian process regression (GPR) methods are advantageous for their ability to quantify the uncertainty of estimated values, which is particularly valuable in practical applications. Nonetheless, GPR methods typically exhibit lower efficiency in high-dimensional spaces and can be computationally complex (Li et al., 2022; Ren and Du, 2023). Another recent approach, with a sole focus on the prediction of the SOH, is presented by Luo et al. (2023), in which the authors introduce the methodology of diffusion models as a promising avenue for SOH prediction.

Despite the widespread application of ML models in battery degradation analysis, comparing these various approaches presents significant challenges. Many studies utilize different datasets, which are often not publicly available due to confidentiality concerns. To address this issue, BatteryML was developed by Zhang et al. (2024b), offering a standardized method for data representation that consolidates and harmonizes all accessible public battery datasets. Additionally, BatteryML establishes clear benchmarks for predicting RUL and includes a range of models, such as linear models, tree-based models, and neural networks, tailored for battery degradation prediction. In a recent study, Zhang et al. (2025) introduced BatLiNet, a CNN-based framework designed to predict

RUL across diverse ageing conditions. Its distinctive feature is an inter-cell learning mechanism that predicts lifetime differences between pairs of cells. When combined with conventional single-cell learning, this approach improves the stability and robustness of RUL predictions.

## 1.4 CONTRIBUTION TO LITERATURE

Our contribution advances the domain of battery degradation prediction in several aspects. First, we introduce *FlowBatt*, a general-purpose generative model trained via flow matching, designed to predict full state-of-health (SOH) trajectories and synthesize realistic degradation curves. Leveraging a scalable diffusion transformer (DiT) backbone, FlowBatt captures the stochastic and nonlinear dynamics of battery aging more effectively than traditional supervised approaches (Section 2). Second, we demonstrate the benefits of flow-based generative modeling by benchmarking FlowBatt against diffusion processes and supervised learning methods. Our results show that FlowBatt achieves competitive performance compared to the state-of-the-art in remaining useful life (RUL) prediction while providing uncertainty-aware and generalizable SOH forecasts (Section 4). Finally, we establish FlowBatt as a scalable foundation model for battery degradation, enabling future extensions and adaptations across diverse energy storage applications (Section 5). All codes and pre-trained models will be made openly available at: `https://github.com/_/FlowBatt`.

## 2 METHODOLOGY

Denoising diffusion probabilistic models (DDPMs) (Ho et al., 2020) are a flexible family of generative models that learn to approximate a data distribution through a gradual noising-denoising process. The forward process is defined as a parameterized Markov chain that progressively corrupts the data with Gaussian noise until the signal is indistinguishable from random noise. The reverse process, also known as the denoising or generative process, is learned by the model and gradually removes noise step by step to reconstruct samples resembling the training data. Training is performed using a variational objective that optimizes a lower bound on the data likelihood across many diffusion steps. Once trained, DDPMs can generate diverse, high-quality samples, which has led to widespread applications in image and audio synthesis (Dhariwal and Nichol, 2021; Ho et al., 2022; Yang et al., 2024), data augmentation (Luzi et al., 2024), and scientific modeling (Bastek et al., 2024; Fürrutter et al., 2024; Li et al., 2024; Zhang et al., 2024a).

Flow matching (Lipman et al., 2023; Liu et al., 2023; Albergo and Vanden-Eijnden, 2023) is a proposed alternative to diffusion training that directly learns continuous-time generative dynamics. Instead of discretizing the forward and reverse processes into thousands of noisy steps as in DDPMs, flow matching estimates an ordinary differential equation (ODE) that smoothly transports a base distribution (e.g., Gaussian noise) into the target data distribution. This approach avoids the stochasticity and step-by-step denoising of DDPMs and enables more efficient training with fewer discretization steps. Moreover, flow matching naturally admits exact likelihood estimation, making it both expressive and computationally attractive for high-dimensional scientific data. In this work, we adopt flow matching to train our generative model of battery degradation.

Transformers have become a powerful backbone for diffusion-based generative models, giving rise to diffusion transformers (DiTs) (Peebles and Xie, 2023). Unlike convolutional UNet backbones, DiTs leverage global self-attention to capture long-range dependencies in the data. Diffusion transformers typically operate on tokenized patches of the input, and use a very powerful conditioning mechanism based on adaptive layer normalization (Perez et al., 2018) for conditioning on auxiliary information such as class labels or text encodings. In this work, the auxiliary information corresponds to the battery's performance during the early cycle life. Our approach employs DiTs directly on raw data representations of battery degradation. This design combines the scalability and expressivity of transformers with the probabilistic training of flow matching to capture the nonlinear, stochastic dynamics of battery aging.

## 2.1 BACKGROUND

Denoising diffusion models (DDMs) learn to transform a simple prior distribution, typically a unit Gaussian $\mathcal{N}(\mathbf{0}, \mathbf{I})$, into an unknown data distribution $q(\boldsymbol{x})$ by gradually denoising samples corrupted with Gaussian noise (Ho et al., 2022). A fixed *forward process* incrementally adds noise to data

$\boldsymbol{x}_0 \sim q(\boldsymbol{x})$ according to a variance schedule, and a neural network is trained to approximate the *reverse process* that removes noise step by step. With the standard parameterization suggested in Ho et al. (2022), the training objective reduces to a denoising score matching loss, where the network $\boldsymbol{\epsilon}_\theta(\boldsymbol{x}_t, t)$ predicts the injected Gaussian noise

$$\mathcal{L}_{\text{simple}}(\theta) := \mathbb{E}_{t, \boldsymbol{x}_0, \boldsymbol{\epsilon}} \left[ \|\boldsymbol{\epsilon}_t - \boldsymbol{\epsilon}_\theta(\boldsymbol{x}_t, t)\|^2 \right]. \tag{1}$$

For conditional modeling, auxiliary information $\boldsymbol{c}$ can be incorporated into the reverse process by conditioning the network $\boldsymbol{\epsilon}_\theta(\boldsymbol{x}_t, t, \boldsymbol{c})$, often enhanced with classifier-free guidance (Ho and Salimans, 2022). While diffusion models achieve strong generative performance, sampling requires simulating many reverse steps, which motivates alternative formulations.

Instead of constructing a discrete forward diffusion process, flow matching (Lipman et al., 2023; Liu et al., 2023) formulates generative modeling as learning a continuous-time transport map between a tractable prior distribution $p(\boldsymbol{x})$, typically $\mathcal{N}(\boldsymbol{0}, \boldsymbol{I})$, and the unknown data distribution $q(\boldsymbol{x})$. The approach defines a family of interpolant distributions $\{p_t(\boldsymbol{x})\}_{t \in [0,1]}$ that smoothly connects the prior $p_0(\boldsymbol{x}) = p(\boldsymbol{x})$ to the target $p_1(\boldsymbol{x}) = q(\boldsymbol{x})$. This interpolation is governed by an ordinary differential equation (ODE) of the form

$$\frac{d\boldsymbol{x}_t}{dt} = \boldsymbol{v}_\theta(\boldsymbol{x}_t, t), \quad \boldsymbol{x}_0 \sim p(\boldsymbol{x}), \tag{2}$$

where $\boldsymbol{v}_\theta$ is a neural network parameterized vector field trained to approximate the unknown probability flow that transports samples from prior to data. To obtain a tractable training objective, one introduces a *target vector field* defined by the time derivative of a prescribed stochastic interpolant $\boldsymbol{x}_t = \alpha(t)\boldsymbol{x}_0 + \beta(t)\boldsymbol{x}_1$ with $(\boldsymbol{x}_0, \boldsymbol{x}_1) \sim p(\boldsymbol{x}) \times q(\boldsymbol{x})$. For example, choosing linear interpolation $\alpha(t) = 1 - t$, $\beta(t) = t$ yields

$$\tilde{\boldsymbol{v}}(\boldsymbol{x}_t, t | \boldsymbol{x}_0, \boldsymbol{x}_1) = \frac{d}{dt} \left[ \alpha(t)\boldsymbol{x}_0 + \beta(t)\boldsymbol{x}_1 \right] = \dot{\alpha}(t)\boldsymbol{x}_0 + \dot{\beta}(t)\boldsymbol{x}_1. \tag{3}$$

The model $\boldsymbol{v}_\theta(\boldsymbol{x}_t, t)$ is then trained to match this target vector field in expectation:

$$\mathcal{L}_{\text{FM}}(\theta) := \mathbb{E}_{t, \boldsymbol{x}_0 \sim p(\boldsymbol{x}), \boldsymbol{x}_1 \sim q(\boldsymbol{x})} \left[ \|\boldsymbol{v}_\theta(\boldsymbol{x}_t, t) - \tilde{\boldsymbol{v}}(\boldsymbol{x}_t, t | \boldsymbol{x}_0, \boldsymbol{x}_1)\|^2 \right]. \tag{4}$$

This training can be interpreted as regressing the dynamics of sample trajectories under the optimal transport plan that pushes the prior into the data distribution. Once trained, the model generates samples by integrating the learned ODE

$$\boldsymbol{x}_1 = \boldsymbol{x}_0 + \int_0^1 \boldsymbol{v}_\theta(\boldsymbol{x}_t, t) \, dt, \quad \boldsymbol{x}_0 \sim p(\boldsymbol{x}), \tag{5}$$

which requires significantly fewer steps than simulating the reverse diffusion process. Moreover, flow matching provides a unifying framework: with appropriate choices of interpolants, it can recover score-based diffusion models as a special case while also supporting more efficient sampling schemes. For conditional generation, the conditioning variable $\boldsymbol{c}$ is incorporated directly into the learned vector field $\boldsymbol{v}_\theta(\boldsymbol{x}_t, t, \boldsymbol{c})$.

## 3 ARCHITECTURE

Figure 1 provides an overview of the FlowBatt architecture and its workflow. The top panel depicts the diffusion-based training stage, where noisy SOH trajectories are progressively denoised through a stack of DiT blocks conditioned on early-cycle capacity matrices. The bottom panels illustrate the auxiliary pathway and the generation stage, where the capacity matrix is first encoded by a CNN into a compact embedding, which is combined with the timestep embedding to modulate the DiT backbone and generate complete SOH trajectories.

**DiT backbone and block design.** FlowBatt instantiates a compact Diffusion Transformer (DiT) for long one-dimensional sequences representing SOH during battery's cycle life. Given a per-cycle signal $\boldsymbol{x} \in \mathbb{R}^T$, we first lift it to a token sequence via a $1 \times 1$ convolution giving $\boldsymbol{h}_0 \in \mathbb{R}^{T \times d}$, where $d$ is the hidden width (number of channels). We add fixed one-dimensional sinusoidal positional embeddings $\boldsymbol{p} \in \mathbb{R}^{T \times d}$ to preserve temporal order. The core of the network is a stack of $N$ identical DiT blocks, each following the Transformer paradigm of a pre-normalized residual connection with a

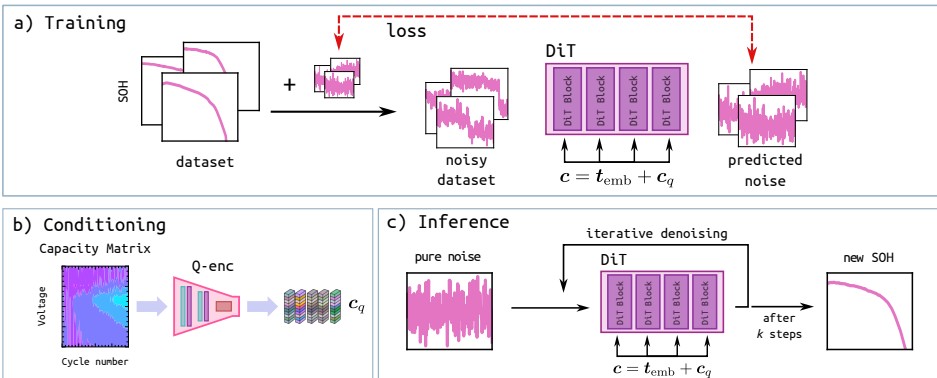

Figure 1: Schematic view of the model architecture. Adapted and modified from the work by Fürrutter et al. (2024), with permission from the authors. Modifications include context-specific changes.

multi-head self-attention (MSA) sublayer, followed by a feed-forward multilayer perceptron (MLP) sublayer. The stack is followed by a final projection layer that applies an adaptive normalization conditioned on the conditioning vector and then maps the hidden states back to the data domain. This ensures that conditioning information influences not only the internal block dynamics but also the final predicted vector field used in flow matching.

**Conditioning mechanism.** In the adaptive layer norm (adaLN)-Zero formulation (Peebles and Xie, 2023), each MSA and MLP sublayers is modulated by per-channel scale and shift parameters $(\gamma, \beta)$ applied to the normalized activations, together with residual scaling factors $\alpha$ applied directly before the residual addition. All modulation parameters are produced from a single global conditioning vector $\boldsymbol{t}_{\mathrm{emb}} + \boldsymbol{c_q}$ that merges (i) a timestep embedding $\boldsymbol{t}_{\mathrm{emb}} = \phi(t)$ and an auxiliary embedding $\boldsymbol{c_q}$ obtained by encoding the capacity matrix corresponding to the early cycle life of the battery. The conditioning vector is passed through a small multilayer perceptron that produces six $d$-dimensional vectors as per-channel modulation parameters. The residual scalars $\alpha$ are initialized to zero, ensuring that every block starts as an identity mapping and that conditioning effects are introduced gradually during training. This approach, has been shown to improve optimization stability and sample quality in large-scale generative modeling (Lipman et al., 2023).

---

**Algorithm 1:** One DiT block with adaLN-Zero conditioning

---

**Input:** Hidden states $\boldsymbol{h} \in \mathbb{R}^{T \times d}$, timestep $t$, auxiliary input $\boldsymbol{Q}$ (capacity matrix).
**Output:** Updated hidden states $\boldsymbol{h}' \in \mathbb{R}^{T \times d}$.

```
/* Step 1:  Form conditioning vector                                        */
t_emb ← φ(t)                                                // timestep embedding
c_q ← CNN(Q)                                               // auxiliary embedding
c ← t_emb + c_q                                      // global conditioning vector
/* Step 2:  Generate modulation parameters                                  */
(γ_attn, β_attn, α_attn, γ_mlp, β_mlp, α_mlp) ← MLP(c)
/* Step 3:  Attention sublayer                                              */
z ← LN(h)                                                          // pre-norm
z ← γ_attn ⊙ z + β_attn                                    // adaLN modulation
h ← h + α_attn ⊙ MSA(z)                                      // residual update
/* Step 4:  MLP sublayer                                                    */
z ← LN(h)                                                          // pre-norm
z ← γ_mlp ⊙ z + β_mlp                                      // adaLN modulation
h' ← h + α_mlp ⊙ MLP(z)                                      // residual update
return h'
```

---

**CNN encoder.** We employ the concept of *capacity matrix* ($\boldsymbol{Q}$) introduced by Attia et al. (2021), as an auxiliary information for conditional generation. The capacity matrix serves as a compact representation of battery electrochemical cycling data, incorporating a series of feature representations. We utilize the capacity matrix obtained from early cycle life of the battery, i.e. the first 20 or 100 cycles similar to Attia et al. (2021); Severson et al. (2019b); Zhang et al. (2024b). This choice is driven by the high costs, time, and effort associated with long-term battery testing. Our goal is to

leverage early life performance data to predict battery degradation and minimize resource expenditure. To encode $Q$ into an embedding ($c_q$), we utilize a CNN encoder (see Figure 1b). We adapt a CNN architecture inspired by Attia et al. (2021); Zhang et al. (2025), which has been used for RUL prediction. This model has been reported to lack robustness with respect to different initializations, however, it provides very accurate predictions in some instances. To enhance the robustness of the model we utilized explainable AI methods, specifically Grad-CAM (gradient-weighted class activation mapping) method (Selvaraju et al., 2017) to better understand how these deep-learning models make predictions for the RUL task. Further, we identified and eliminated potential modes of failure. Results for this analysis are reported in Appendix A.2. The final architecture consists of two convolutional layers with leaky rectified linear unit (LeakyReLU) activations, each followed by average pooling to reduce spatial dimensions. The resulting feature map is projected with a final convolution, activated, and flattened into a $d$-dimensional vector $c_q$. During training, classifier-free guidance can be employed by randomly dropping the embedding, allowing the model to learn both conditional and unconditional generations. This CNN encoder effectively captures the relevant features from the capacity matrix, enabling FlowBatt to generate accurate and context-aware SOH trajectories.

In summary, our generative models trained via flow matching and diffusion processes combine a DiT backbone with adaLN-Zero conditioning and a CNN-based encoder for auxiliary early-life data, yielding a scalable and uncertainty-aware generative framework. By integrating timestep information with early-cycle capacity matrices, the model learns to generate realistic SOH trajectories that reflect both temporal dynamics and usage context. From these SOH trajectories, metrics such as RUL can be directly computed by applying practical thresholds (e.g., 80% of nominal capacity). This design positions FlowBatt as a flexible model for battery health prediction, capable of both probabilistic modeling of aging behavior and data-driven synthesis. Figure 2 illustrates the denoising trajectories of a test sample from the MATR-1 dataset using flow matching and diffusion processes. It can be observed that the flow matching approach achieves faster denoising compared to the diffusion process. Note that diffusion requires 1000 reverse steps, while in flow matching we solve the ODE over [0,1].

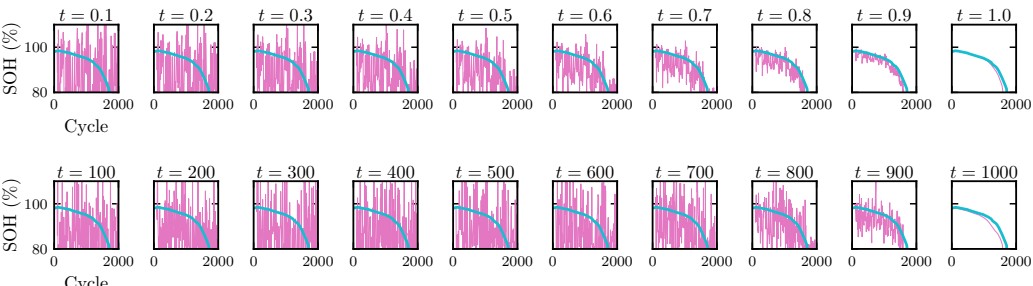

Figure 2: Comparison of denoising trajectories for a test sample from the MATR-1 dataset using flow matching (top) and the diffusion process (bottom). The pink line shows the generated sample and the cyan line shows the reference for a given capacity matrix.

## 3.1 DATA

We conduct a comprehensive evaluation based on several publicly available datasets curated in BatteryML (Zhang et al., 2024b). Specifically, we include CALCE (He et al., 2011; Xing et al., 2013), HUST (Ma et al., 2022), MATR (Hong et al., 2020; Severson et al., 2019b), RWTH (Li et al., 2021), SNL (Preger et al., 2020), and UL_PUR (Juarez-Robles et al., 2020; 2021), covering a range of chemistries including lithium iron phosphate (LFP), lithium cobalt oxide (LCO), nickel manganese cobalt oxide (NMC), nickel cobalt aluminum oxide (NCA), and a combination of NMC and LCO (NMC_LCO). These datasets span diverse ageing conditions, including differences in capacities, voltages, ambient temperatures, state of charge (SOC), and RUL ranges and provide a broad spectrum of degradation behaviors.

In this study, we follow the benchmark protocol of BatLiNet (Zhang et al., 2025), which derives five evaluation datasets under different ageing conditions. Specifically, BatLiNet defines two MATR evaluation sets (MATR-1 and MATR-2) to maintain comparability with existing models. In addition,

the HUST dataset, which uses LFP cells similar to MATR, is included to evaluate model adaptation to different cycling protocols. Furthermore, BatLiNet constructs a MIX dataset by aggregating all remaining public datasets, capturing diverse ageing conditions such as different ambient temperatures, packing structures, and cathode chemistries (NMC, LCO, and NCA). From the MIX dataset, two prediction tasks are defined: MIX-100, which evaluates the prediction of the 80% end-of-life point using only the first 100 cycles, and MIX-20, a more challenging setup requiring the prediction of the number of cycles until 90% capacity using only the first 20 cycles. As in Zhang et al. (2025), batteries that reached end-of-life prematurely during the early cycles are excluded from the analysis. Further details on datasets and preprocessing are provided in Appendix A.1 and Zhang et al. (2025; 2024b).

## 4 RESULTS AND DISCUSSION

Our generative models can be utilized for RUL prediction, SOH estimation, and SOH synthesis. It should be noted that we refer to generating new SOH curves for data augmentation as SOH synthesis. SOH and RUL predictions are two common tasks in managing inevitable capacity fade, widely discussed in the literature. While both tasks aim to predict capacity fade, they utilize different data representations. In this section, we conduct experiments based on the BatLiNet benchmark tests (Zhang et al., 2025) and compare the results. Further detailed analysis and results are included in Appendix A.4.

### 4.1 RUL PREDICTION

We evaluate three different models trained with supervised learning, diffusion processes, and flow matching, referred to as Transformer, DiffBatt, and FlowBatt, respectively. All models directly predict SOH trajectories from the input capacity matrix, from which the RUL is computed as the cycle index at which the SOH drops below a specified threshold (e.g., 80% or 90%). For the probabilistic models (DiffBatt and FlowBatt), a single model is trained and evaluated by generating ten SOH trajectories from ten independent noise realizations for each input, thereby quantifying predictive variability. For the deterministic baseline (Transformer), ten models with different random initializations are trained, and we report the mean and standard deviation of their predictions.

Table 1: Comparison with baseline methods for the RUL prediction task. Best results based on RMSE are shown in bold, and second-best results are underlined. For the models introduced in this study, we report the mean and standard deviation over ten random seeds.

| Method | MATR-1 | | MATR-2 | | HUST | | MIX-100 | | MIX-20 | |
| | RMSE | MAPE (%) | RMSE | MAPE (%) | RMSE | MAPE (%) | RMSE | MAPE (%) | RMSE | MAPE (%) |
|---|---|---|---|---|---|---|---|---|---|---|
| Training Mean | 399 | 28 | 511 | 36 | 420 | 18 | 573 | 59 | 593 | 102 |
| "Variance" Model (Severson et al., 2019a) | 138 | 15 | 196 | 12 | 398 | 17 | 521 | 39 | 601 | 95 |
| "Discharge" Model (Severson et al., 2019a) | 86 | 8 | _173_ | _11_ | 322 | 14 | 1743 | 47 | >2000 | >100 |
| "Full" Model (Severson et al., 2019a) | 100 | 11 | 214 | 12 | 335 | 14 | 331 | 22 | 441 | 53 |
| Ridge Regression (Attia et al., 2021) | 125 | 13 | 188 | 11 | 1047 | 36 | 395 | 30 | 806 | 150 |
| PCR (Attia et al., 2021) | 100 | 11 | 176 | 11 | 435 | 19 | 384 | 28 | 701 | 78 |
| PLSR (Attia et al., 2021) | 97 | 10 | 193 | 11 | 431 | 18 | 371 | 26 | 543 | 77 |
| SVM Regression (Zhang et al., 2025) | 140 | 15 | 300 | 18 | 344 | 16 | 257 | 18 | 438 | 46 |
| Random Forest (Attia et al., 2021) | 140 | 15 | 202 | 11 | 348 | 16 | 211 | 14 | 288 | 31 |
| MLP (Attia et al., 2021) | 162±7 | 12±0 | 207±4 | 11±0 | 444±5 | 18±1 | 455±37 | 27±1 | 532±25 | 61±6 |
| LSTM | 123±11 | 12±2 | 226±36 | 14±2 | 442±32 | 20±1 | 266±11 | 15±1 | 417±62 | 37±7 |
| CNN (Attia et al., 2021) | 115±96 | 9±6 | 237±107 | 17±8 | 445±35 | 21±1 | 261±38 | 15±1 | 785±132 | 41±4 |
| BatLiNet (Zhang et al., 2025) | **59±2** | **6±0** | **163±12** | **11±1** | _264±9_ | 10±1 | **158±7** | **10±0** | **201±18** | **18±1** |
| Transformer (ours) | 85±10 | 8±1 | 226±39 | 14±2 | 295±18 | 14±1 | 221±34 | 15±3 | 312±40 | 27±5 |
| DiffBatt (ours) | 68±3 | 6±0 | 202±5 | 16±0 | 282±16 | 10±1 | 218±17 | 13±1 | 295±24 | 30±2 |
| FlowBatt (ours) | _67±2_ | 6±0 | 175±4 | 11±0 | **222±12** | **9±0** | _179±6_ | 11±0 | _229±12_ | 18±1 |

The results for the RUL prediction task are summarized in Table 1. This table illustrates the performance of the models in the RUL task using root-mean-squared error (RMSE) and mean-absolute-percentage error (MAPE) as the evaluation metrics. RMSE is suitable for the RUL task since it represents the error based on an average number of cycles in which the predicted RUL differs from the reference. Overall, FlowBatt achieves strong and consistent performance across all datasets. It obtains the best results on HUST ($222 \pm 12$ cycles, 9% MAPE), while also reaching the second-best on MIX-100 ($179 \pm 6$ cycles, 11% MAPE), MIX-20 ($229 \pm 12$ cycles, 18% MAPE), and MATR-1 ($67 \pm 2$ cycles, 6% MAPE), closely behind BatLiNet. These results highlight FlowBatt's robustness in capturing ageing dynamics across diverse conditions and its capability in down stream tasks, achieving competitive performance compared to the state-of-the-art BatLiNet model (Zhang et al.,

2025), which is specifically designed for RUL prediction. Notably, FlowBatt is the only model besides BatLiNet that consistently ranks within the top three across all datasets, demonstrating superior generalizability. DiffBatt also demonstrates strong predictive capability, achieving lower RMSE than many classical and deep-learning baselines, for example $202 \pm 5$ on MATR-2 and $218 \pm 17$ on MIX-100, however, it generally lags behind FlowBatt. The supervised Transformer baseline performs competitively on simpler datasets such as MATR-1 ($85 \pm 10$) but exhibits larger errors and variance on more challenging setups, indicating limited generalization capacity.

A broader comparison confirms that FlowBatt narrows the gap to or performs better than BatLiNet (Zhang et al., 2025), the current state-of-the-art approach. On the MATR-2 dataset, FlowBatt achieves an RMSE of $175 \pm 4$, only slightly higher than BatLiNet ($163 \pm 12$) but considerably outperforming both DiffBatt ($202 \pm 5$) and Transformer ($226 \pm 39$). DiffBatt remains competitive and provides a useful comparison point for diffusion-based generative modeling, while the Transformer baseline illustrates the limitations of purely supervised approaches when faced with heterogeneous ageing conditions. Taken together, these results confirm that flow-matching-based generative modeling offers a competitive and scalable alternative to both supervised and diffusion-based methods for battery degradation.

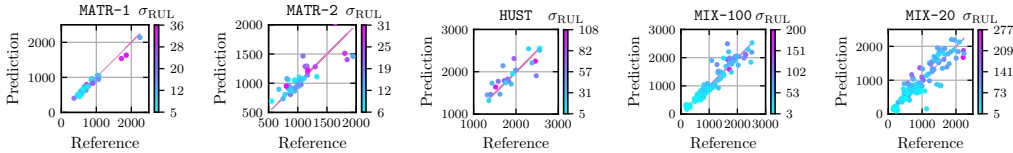

Figure 3: Results obtained from FlowBatt for RUL prediction. Colorbar shows the standard deviation $\sigma_{\text{RUL}}$ of the predicted RUL from ten generated SOH trajectories per sample.

Figure 3 presents the RUL prediction results for all test samples across datasets using the FlowBatt model. FlowBatt provides uncertainty estimates by reporting the standard deviation of RUL values computed from ten generated trajectories per sample. The results suggest a general tendency for samples with larger prediction errors to also exhibit higher predictive variance, indicating that FlowBatt can capture uncertainty in more challenging prediction cases.

## 4.2 SOH PREDICTION

In practical applications, estimating a battery's SOH requires predicting the current discharge capacity under standardized conditions using reference performance tests and historical cycling data. However, the discrepancy between real-world battery usage and these standardized conditions poses significant challenges in obtaining precise ground-truth labels, complicating accurate SOH prediction. Consequently, developing a robust benchmark test for SOH prediction remains an ongoing effort within the research community. Therefore, we benchmark our SOH prediction results by approximating its theoretical definition (capacity relative to the nominal capacity) using data obtained from controlled cycling experiments. Results are reported in Table 2. We employ the same datasets in our RUL prediction tasks for the SOH prediction experiments, which allows for reproducibility due to the clear data splits. For each degradation curve, we compute the error as the RMSE between the predicted and reference trajectories up to the end-of-life (EOL) point, padding the remainder of the sequence with the final observed value. Results are reported as the mean RMSE across all test samples. Since SOH is expressed as a percentage of the nominal capacity, the mean RMSE is also given in percentage units. To ensure reproducibility, a detailed discussion on experimental setup and error metrics is included in Appendix A.4. The supervised Transformer baseline provides a useful point of comparison, achieving reasonable performance on some datasets, such as MATR-1 (1.09±0.07) and MIX-20 (1.42±0.10), but showing considerably higher errors on more challenging datasets such as HUST (3.18±0.16). This highlights the limited generalization ability of purely supervised models across diverse ageing conditions. DiffBatt improves upon the Transformer across all datasets, reducing the RMSE and demonstrating the benefits of probabilistic generative modeling. For example, DiffBatt achieves an RMSE of 0.92±0.04 on MATR-1 and 1.15±0.08 on MIX-20, both lower than the Transformer baseline. FlowBatt further advances accuracy and generalizability, consistently outperforming both Transformer and DiffBatt. It achieves the best results on all datasets, with an RMSE of 0.87±0.02 on MATR-1, 1.77±0.04 on MATR-2, and 2.10±0.12 on HUST, showing robustness across different

Table 2: Results obtained for SOH prediction task corresponding to the mean RMSE and the standard deviation across ten predictions.

| Method | MATR-1 | MATR-2 | HUST | MIX-100 | MIX-20 |
|---|---|---|---|---|---|
| Transformer | 1.09±0.07 | 2.25±0.22 | 3.18±0.16 | 1.76±0.06 | 1.42±0.10 |
| DiffBatt | 0.92±0.04 | 1.85±0.03 | 2.44±0.14 | 1.71±0.09 | 1.15±0.08 |
| FlowBatt | **0.87±0.02** | **1.77±0.04** | **2.10±0.12** | **1.41±0.02** | **0.88±0.03** |

battery chemistries and operating conditions. Particularly on the heterogeneous MIX-100 and MIX-20 datasets, FlowBatt achieves significant improvements, lowering the RMSE to 1.41±0.02 and 0.88±0.03, respectively. These results demonstrate that flow matching provides a more scalable and reliable approach for SOH trajectory prediction, yielding both higher accuracy and stronger generalization than supervised or diffusion-based baselines.

## 5 TOWARDS A FOUNDATION MODEL FOR BATTERY DEGRADATION

Foundation models are large-scale, general-purpose AI models pre-trained on broad datasets and adaptable to downstream tasks with minimal retraining (Bommasani et al., 2022). By training on diverse battery datasets and demonstrating strong generalization across prediction tasks, FlowBatt offers a promising pathway toward foundational modeling of battery degradation. It captures complex ageing dynamics across chemistries and operating conditions, showing high expressivity and robust generalization. The architecture naturally supports multimodal conditioning: in addition to early-cycle capacity matrices, inputs such as temperature, current profiles, or environmental conditions can be incorporated, enabling rapid adaptation to new chemistries or protocols with limited fine-tuning. FlowBatt also scales efficiently to large datasets through its flow-matching transformer backbone and can synthesize realistic degradation curves to augment scarce datasets, thereby improving robustness and transferability of downstream models.

## 6 CONCLUSIONS AND OUTLOOK

Tackling battery degradation is a major hurdle in advancing green technologies and sustainable energy solutions. Accurately predicting battery capacity loss remains particularly challenging due to its intricate and complex nature. To address this issue, we present FlowBatt, a novel general-purpose model for predicting and synthesizing battery degradation patterns based on flow matching and diffusion transformers. While the majority of the literature is dedicated to RUL predictions, we present FlowBatt as a generative model for probablistic modeling of full SOH trajectories. FlowBatt functions as both a probabilistic model to capture the inherent uncertainties in aging processes and a generative model to simulate and predict battery degradation over time.

We evaluate the performance of Transformer, DiffBatt, and FlowBatt across two different tasks, i.e., RUL prediction and SOH prediction. In the RUL prediction task, FlowBatt achieves competitive performance, closely approaching the state-of-the-art BatLiNet model while consistently outperforming DiffBatt and the supervised Transformer baseline. It achieves the best results on HUST with RMSE of 222±12 cycles and ranks among the top three models across all the datasets, performing closely behind BatLiNet. These results illustrate FlowBatt's efficacy in learning from and generalizing across diverse data sources. In the SOH prediction task, FlowBatt achieves the lowest errors across all datasets, highlighting the advantages of flow matching over both diffusion-based training and supervised learning.

We believe that by training on several diverse battery datasets and demonstrating strong generalizability and robustness across various tasks, FlowBatt offers a promising pathway toward developing a foundational model for battery degradation. To support a deeper understanding of degradation mechanisms and to derive countermeasures in battery design, production, and formation, the data-driven FlowBatt model may be linked with physics-based approaches or extended to account for variations in design and process parameters. Beyond the technical contributions, we hope this work motivates stronger engagement from industry and government stakeholders in curating real-world field data and making them publicly available, thereby accelerating progress in this critical area of research.

## USAGE OF LARGE LANGUAGE MODELS

Large language models were used solely for language editing, including linguistic condensation and checks for grammar and sentence structure. All content was reviewed and approved by the authors.

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

# A   APPENDIX

## A.1   DATASETS

Battery degradation curves utilized for training and testing the models are depicted in Figure 4 for each cell chemistry. A brief summary of the datasets included in BatteryML (Zhang et al., 2024b) is presented here.

The CALCE dataset includes full lifecycle data from 13 batteries with an LCO cathode. Each battery has a nominal capacity of 1100 mAh. They were all charged using a constant current/constant voltage protocol: 0.5C current until reaching 4.2V, maintaining 4.2V until the current dropped below 0.05A, and a cutoff voltage of 2.7V (Xing et al., 2013; He et al., 2011).

The MATR dataset, provided by Severson et al. (2019b) and Hong et al. (2020), is one of the largest public datasets containing 180 commercial 18650 LFP batteries. These batteries, cycled at a forced convection temperature chamber of 30°C, have a nominal capacity of 1.1 Ah and a nominal voltage of 3.3V. The dataset comprises three subsets: MATR-1, MATR-2 (Severson et al., 2019b), and CLO (Hong et al., 2020), all categorized due to distinct measurement batches.

The HUST dataset includes 77 LFP batteries, similar to those in the MATR dataset. These batteries followed an identical charging protocol with varying multi-stage discharge protocols, all conducted at a constant temperature of 30°C (Ma et al., 2022).

The HNEI dataset contains 14 commercial 18650 cells with a graphite anode and a blended NMC and LCO cathode. These cells were cycled at 1.5C to 100% depth of discharge for over 1000 cycles at room temperature (Devie et al., 2018).

The SNL dataset includes 61 commercial 18650 cells (NCA, NMC, and LFP), cycled to 80% capacity. The study evaluates the impact of temperature, depth of discharge, and discharge current on long-term degradation (Preger et al., 2020).

The UL_PUR dataset comprises 10 commercial pouch cells with a graphite negative electrode and an NCA cathode. These cells were cycled at 1C between 2.7V and 4.2V, equivalent to 0-100% state of charge (SOC), at room temperature until reaching 10-20% capacity fade. Additionally, modules were cycled at C/2 between 13.7V and 21.0V until 20% capacity fade (Juarez-Robles et al., 2020; 2021).

The RWTH dataset contains data from 48 lithium-ion battery cells aged under identical conditions. These cells feature a carbon anode and an NMC cathode (Li et al., 2021). The cells were cycled at a constant ambient temperature of 25°C. Each cycle involved a 30-minute discharge phase down to 3.5V and a 30-minute charge phase up to 3.9V, with the currents capped at a maximum of 4A. This resulted in cycles between approximately 20% and 80% state of charge.

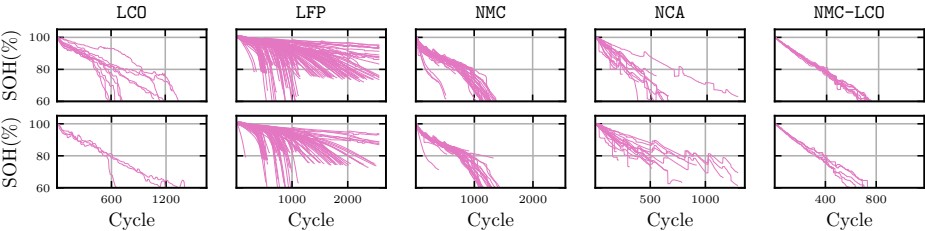

Figure 4: Train (up) and test (bottom) samples for each cell chemistry. The data is scaled using the SOH of the first cycle.

## A.2   EXPLAINING CNN PERFORMANCE VIA GRAD-CAM

For the RUL prediction task, we implement a CNN model based on the architecture proposed in Attia et al. (2021); Zhang et al. (2024b; 2025). The model consists of convolutional layers with ReLU activation functions, followed by a fully connected layer. The model is trained using the Adam optimizer with a learning rate of 0.001 and a batch size of 128 (this is equal to full-batch training for MATR-1 dataset) for 1000 epochs. The model is trained via mean-squared error (MSE) as the

loss function. We train ten models with different initialization seeds and report the RMSE for each model on the test dataset. The results are summarized in Table 3 for different seeds. As observed in the table, the CNN model exhibits high variability in performance across different seeds, with RMSE values ranging from 60 to 306 cycles. This variability indicates that the model's performance is sensitive to initialization and may not consistently capture the underlying degradation patterns. This lack of robustness has been also reported in Attia et al. (2021); Zhang et al. (2024b; 2025). To

Table 3: RUL prediction results obtained from CNN models with different initialization seeds for `MATR-1` dataset.

| Seed | 0 | 1 | 2 | 3 | 4 | 5 | 6 | 7 | 8 | 9 |
|---|---|---|---|---|---|---|---|---|---|---|
| Benchmark CNN (Zhang et al., 2024b) | 76 | 67 | 64 | 74 | 60 | 82 | 65 | 79 | 367 | 78 |
| CNN (Replicated) | 71 | 77 | 83 | 306 | 79 | 60 | 78 | 62 | 88 | 64 |
| XAI-CNN (Ours) | 57 | 54 | 54 | 72 | 66 | 64 | 65 | 60 | 59 | 61 |

identify the reason behinde the high variability in performance, we employ Grad-CAM (Selvaraju et al., 2017) to visualize the regions of the input capacity matrix that the CNN model focuses on when making predictions. Figure 5(a) shows one sample of the capacity matrix from the `MATR-1` train dataset, while Figure 5(b) displays all test samples of the `MATR-1` dataset. The heatmaps in Figure 6 illustrate the Grad-CAM results for the best and worst-performing CNN models based on RMSE. The heatmaps obtained from the worst model indicate that the gradients for some test samples are zero, suggestiong that the model fails to perform meaningful predictions for these samples. Same observations for the training samples indicate that the model fails to learn from those training samples. Moreover, the model tends to focus on early cycles, which may not provide sufficient information for accurate RUL prediction. In contrast, the heatmaps for the best-performing model show that the model focuses on a broader range of cycles, including later ones that are more indicative of degradation trends.

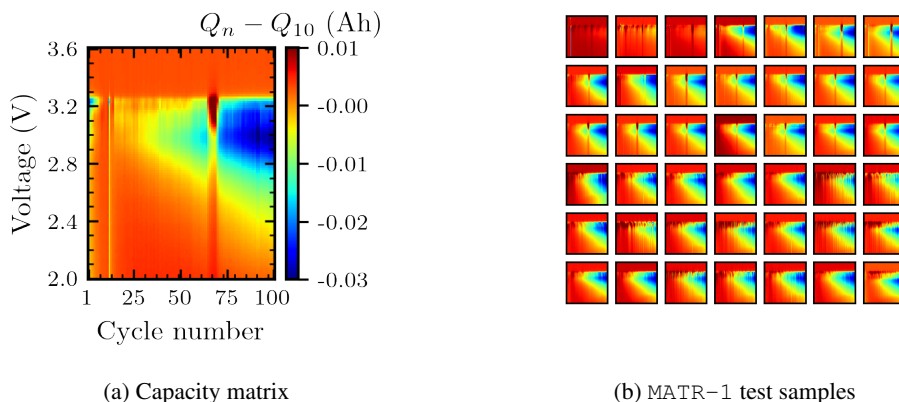

(a) Capacity matrix

(b) `MATR-1` test samples

Figure 5: (a) One sample of capacity matrix from `MATR-1` train dataset. (b) All test samples of `MATR-1` dataset.

To enhance the robustness and accuracy of the CNN model, we provide a simple remedy by replacing the ReLU activation functions with Leaky ReLU activation functions with negative slope equal to 0.3. This choice is due to the realization that the ReLU activation function can lead to dead neurons, especially when the input data contains negative values or when the weights are initialized poorly. Dead neurons do not contribute to learning, which can significantly impact the model's performance and robustness. Leaky ReLU allows a small, non-zero gradient when the unit is not active, which helps to keep the neurons alive and ensures that they continue to learn during training. This modification helps to mitigate the issue of dead neurons and improves the overall robustness of the CNN model. We refer to this modified model as XAI-CNN. The XAI-CNN model is trained using the same hyperparameters as the CNN model. The RMSE results for different seeds are summarized in Table 3. This model demonstrates significantly improved robustness, with RMSE values ranging from 54 to 72 cycles across different seeds, and overal provides very accurate RUL

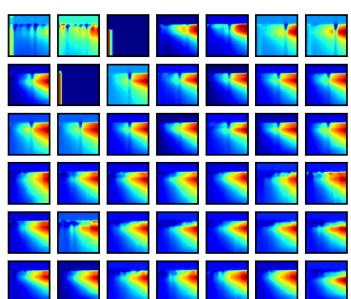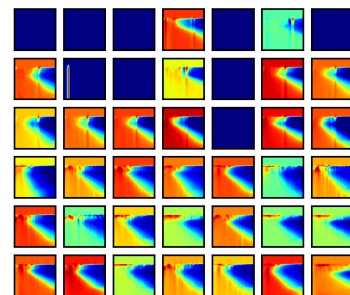

(a) Grad-CAM heatmaps for best model

(b) Grad-CAM heatmaps for worst model

Figure 6: Grad-CAM heatmaps for CNN model proposed in Attia et al. (2021); Zhang et al. (2024b).

predictions with RMSE of 61±5 for `MATR-1` dataset. The Grad-CAM heatmaps for the best and worst-performing XAI-CNN models are shown in Figure 7. The heatmaps indicate that the XAI-CNN model consistently focuses on relevant regions of the capacity matrix across all test samples, suggesting that it effectively captures important degradation patterns. This improvement in focus likely contributes to the enhanced performance and robustness of the XAI-CNN model compared to the original CNN architecture.

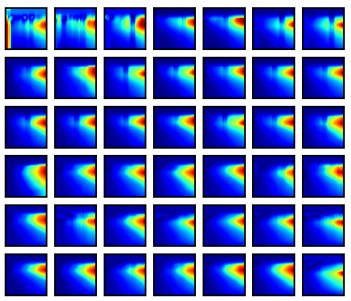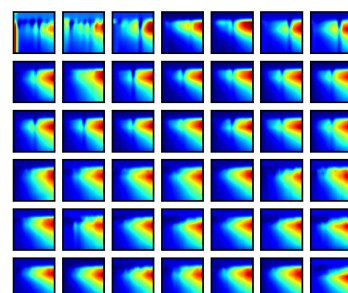

(a) Grad-CAM heatmaps for best model

(b) Grad-CAM heatmaps for worst model

Figure 7: Grad-CAM heatmaps for XAI-CNN model.

We use the XAI-CNN model as the CNN encoder in FlowBatt and DiffBatt. The final linear layer of the XAI-CNN model is removed, and the output from the last convolutional layer is flattened and passed to a linear layer to match the dimension of the condition vector in FlowBatt and DiffBatt.

## A.3 HYPERPARAMETERS

Table 4 summarizes the hyperparameters used for training our FlowBatt model. For implementing flow matching, we build upon the official `flow_matching` library from Facebook Research (Lipman et al., 2024). For diffusion-related components, including optimizers and schedulers, we rely on the Hugging Face `diffusers` package von Platen et al. (2022).

For model development, we preprocess the SOH trajectories; we limit each trajectory to a maximum of 2560 charge-discharge cycles and subsample the SOH values every ten cycles. Furthermore, we truncate the trajectory once the SOH drops below 70%, since this threshold is typically considered below the end-of-life criterion for many applications. If a trajectory reaches end-of-life before 2560 cycles, we pad the remaining part of the sequence with a constant SOH value of 70%. As a result, all SOH trajectories are represented as fixed-length sequences of 256 steps, which defines the input dimension of our model.

Table 4: Training and model hyperparameters.

| Category | Major hyperparameters |
| --- | --- |
| Training | Epochs: 10000
Initial learning rate: $1 \times 10^{-3}$
Optimizer: AdamW
Learning rate scheduler: Cosine schedule with 100 warmup steps |
| Model (DiT) | Sequence lenght: 256
Capacity matrix shape: (1,100,100)
Number of DiT blocks: 4
Channels (embedding dim): 16
Attention heads: 1
MLP ratio: 4.0
Class dropout probability: 0.0 |
| Flow matching | Scheduler: Conditional optimal transport schedule |

## A.4 SOH PREDICTION

For prediction tasks we generate ten samples for each input capacity matrix. The capacity matrix is constructed from the first 100 or 20 cycles. The RMSE for an SOH sample $j$ is computed as

$$\text{RMSE}_j = \sqrt{\frac{1}{n_j} \sum_{i=1}^{n_j} (\tilde{y}_i - y_i)^2} \tag{6}$$

where $\tilde{y}$ and $y$ represent the predicted and the reference SOH in percentage, respectively, $i$ denotes the cycle number and $n_j$ is the cycle number at which the predicted SOH reaches the EOL. Further, we report the mean RMSE across all the test samples as the RMSE for the dataset. We pad both the reference and predicted sequences with the last value to ensure they have the same length. The mean and standard deviation of the RMSE are computed over ten generated samples per each input capacity matrix.

Figure 8 illustrates the predicted SOH versus the reference SOH for all test samples in the MIX-100 dataset. The results demonstrate that FlowBatt effectively captures various degradation dynamics and accurately predicts SOH for the majority of test samples and highlights FlowBatt's ability to generalize across different battery chemistries and operational conditions present in the MIX-100 dataset. This capability is essential for developing reliable battery health monitoring systems that can adapt to diverse usage patterns and environmental factors.

Figure 9 and Figure 10 show the predicted SOH against the reference SOH for all test samples in the MIX-100 dataset obtained from DiffBatt and Transformer, respectively. Compared to FlowBatt, DiffBatt shows a higher deviation from the reference for several test samples. The Transformer model shows a higher deviation from the reference for most of the test samples compared to both FlowBatt and DiffBatt. These results further highlight the advantages of flow matching over both diffusion-based training and supervised learning.

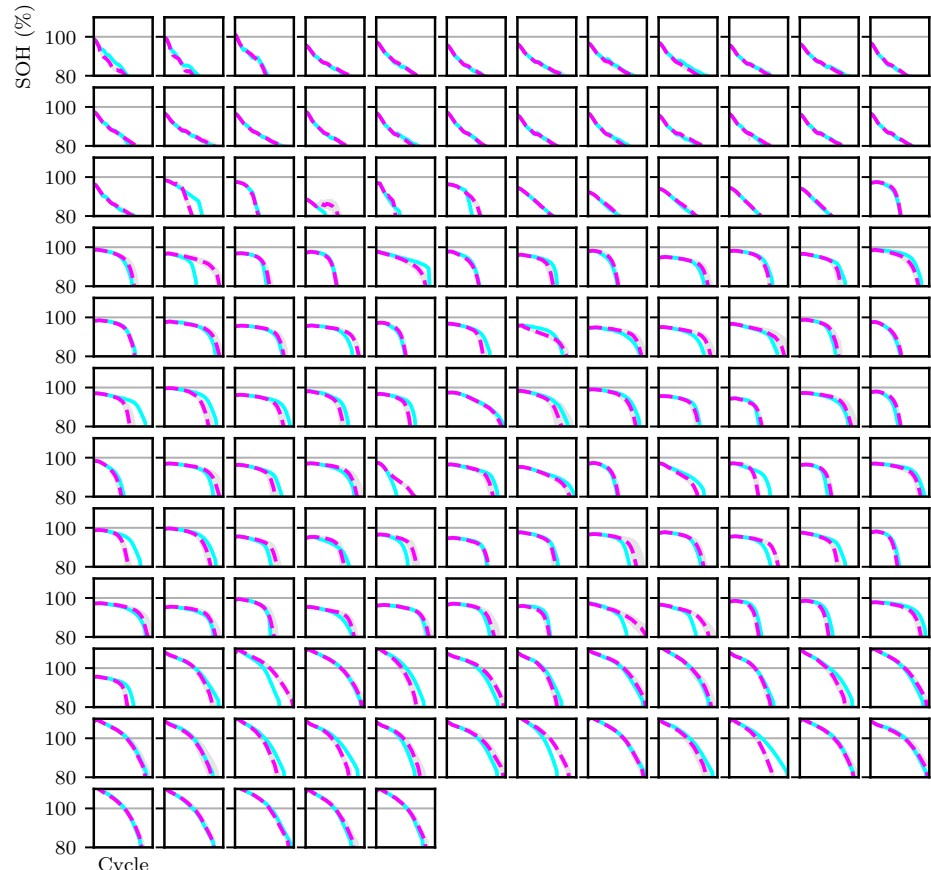

Figure 8: SOH predictions obtained from FlowBatt against reference for all the test samples of MIX-100 dataset. The pink dashed line shows the prediction and the cyan solid line shows the reference. The gray area shows the prediction uncertainty computed from ten generated samples.

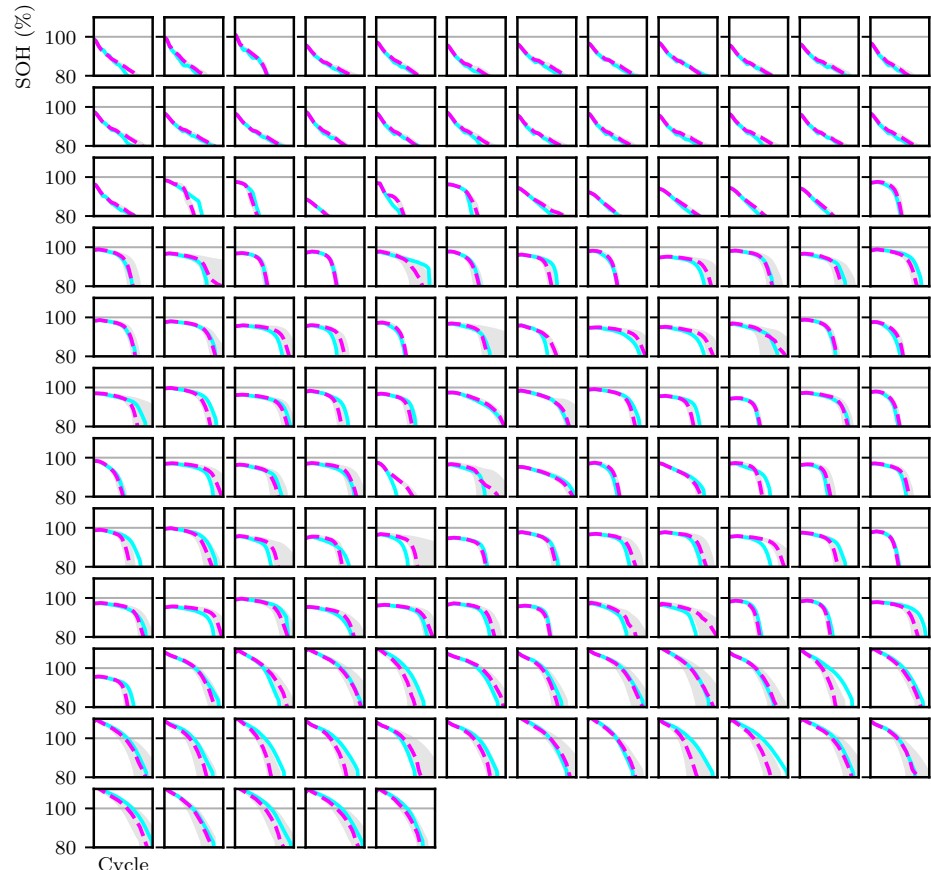

Figure 9: SOH predictions obtained from DiffBatt against reference for all the test samples of MIX-100 dataset. The pink dashed line shows the prediction and the cyan solid line shows the reference. The gray area shows the prediction uncertainty computed from ten generated samples.

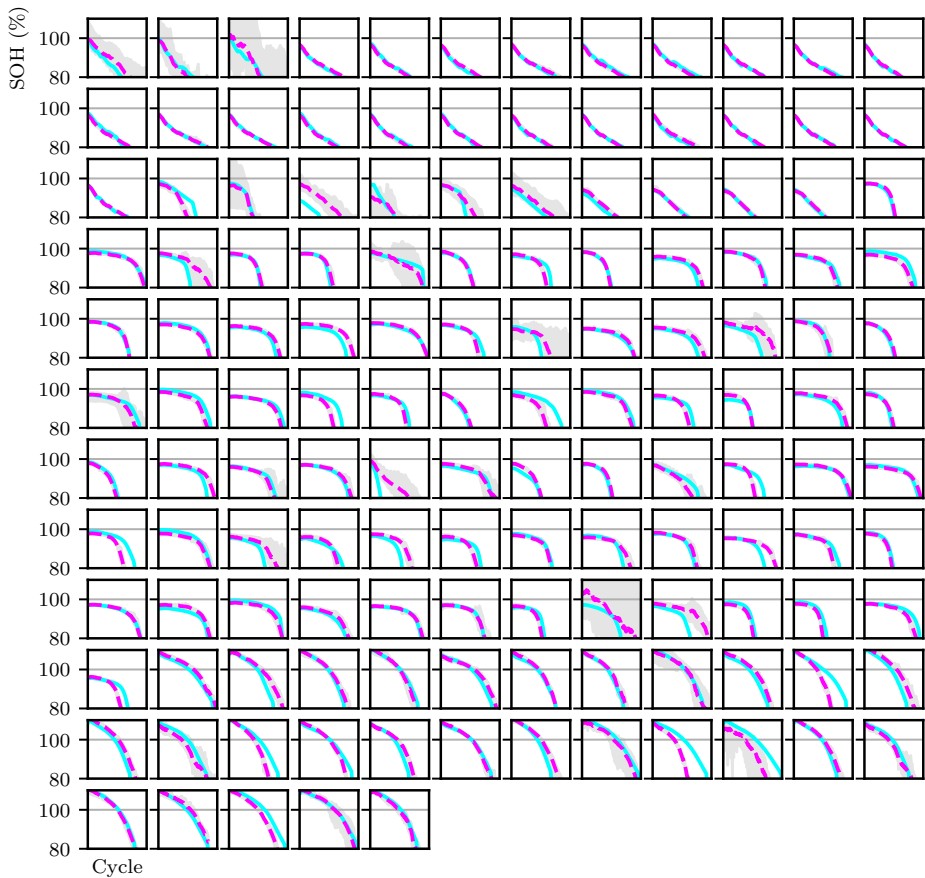

Figure 10: SOH predictions obtained from Transformer model against reference for all the test samples of MIX-100 dataset. The pink dashed line shows the prediction and the cyan solid line shows the reference. The gray area shows the prediction uncertainty computed from ten generated samples.

