# OpenReview forum: "Generative Modeling of Battery Degradation with Flow Matching and Diffusion Transformers"
_ICLR.cc/2026/Conference — Submitted to ICLR 2026_

### Official Review · Reviewer_YzZ2 · 2025-10-30

**Soundness:** 2
**Presentation:** 2
**Contribution:** 2
**Rating:** 4
**Confidence:** 3

**Summary:**

This paper proposes FlowBatt, a conditional generative model for RUL and SOH prediction. The idea is to capture the stochastic and nonlinear dynamics of battery aging by modeling SOH degradation curves with flow-matching. Experiment show that FlowBatt achieves RUL prediction accuracy comparable to SOTA task-specific models, and higher SOH prediction accuracy compared to “Transformer” and diffusion baselines.

**Strengths:**

1. The idea of introducing generative model as a foundation model to predict RUL and SOH is innovative.
2. The proposed method exhibits promising performance on both RUL and SOH tasks on diverse datasets, showcasing its robustness across diverse battery conditions and tasks.
3. Using GradCAM, the author discovered the reason why training of typical CNN on SOH data is unstable, and solves the problem with a non-trivial trick: to replace ReLU with Leaky-ReLU.

**Weaknesses:**

1. **Lack of Novelty**: The proposed method seems to be a direct application of DiT-based flow-matching model on SOH synthesis. The only modification is introducing a CNN network with Leaky-ReLU to encode capacity matrix Q as condition.
2. **Writing**: The structure of the paper is a bit confusing: A considerably large part of the main paper focuses on the background and application of SOH/RUL prediction. However, I suggest including a formal task definition of SOH/RUL definition (as in Appendix A.4), as well as detailed experimental settings (e.g., size of training and evaluation set, size of capacity matrix used for each evaluation set, details on baseline methods, etc.).

**Questions:**

1. Why can DiT-based flow-matching model better deal with the "complex interplay of aging and cycling behaviors"? It would be better if the authors clearly state the core research problem, and motivation to utilize DiT-based flow-matching to solve that problem.
2. Why aren't classical and deep-learning baselines like Random Forests and CNN included in the SOH prediction task?
3. I noticed a large difference between performance of Transformer baseline reported in Table 1 and that reported in [1]. Why? How is the supervised Transformer baseline trained? Is it autoregressive or non-autoregressive? Does it follows the same architecture as DiT?
4. How much data is used to train/evaluate FlowBatt (calculated by number of SOH curves)? This may be important since the meaning of "scalable" is ambiguous. To confirm scalability, experiments to train larger model on large datasets should be conducted.
5. The claim that "the architecture naturally supports multimodal conditioning" may need experimental evidence, since capacity matrices itself are SOH data, whereas there is a domain gap between SOH data and other conditions like current profiles.

**References**

[1] Zhang, Han, et al. "BatteryML: An Open-source Platform for Machine Learning on Battery Degradation." *The Twelfth International Conference on Learning Representations*.

---

### Official Review · Reviewer_ZjNP · 2025-10-31

**Soundness:** 2
**Presentation:** 3
**Contribution:** 2
**Rating:** 4
**Confidence:** 3

**Summary:**

This paper proposes FlowBatt, a generative modeling framework for battery degradation trajectory prediction based on flow matching and a Diffusion Transformer (DiT) architecture. The approach aims to forecast State of Health (SOH) and Remaining Useful Life (RUL) while capturing uncertainty through probabilistic trajectory generation. The model is evaluated on multiple datasets from the BatLiNet benchmark and compared with both diffusion-based and conventional deep learning baselines. The authors argue that FlowBatt improves uncertainty estimation and interpretability while maintaining competitive accuracy.

**Strengths:**

- Introduces flow matching to the battery degradation domain—a relatively unexplored direction. The method’s framing of SOH trajectory generation as a conditional generative process is conceptually novel.

- The experiments are well organized and conducted on diverse, standardized datasets (MATR, HUST, MIX). The paper provides comparisons with diffusion-based (DiffBatt) and supervised baselines under the same protocol.

- The manuscript is clearly structured, with detailed architecture and training explanations. Figures and tables are well presented.

**Weaknesses:**

- Overstated novelty and significance:
The “foundation model” claim is not sufficiently supported given the modest model size and limited cross-domain transfer evaluation. The paper should temper its claims or demonstrate scaling behavior and transferability.

- Limited empirical advantage:
FlowBatt’s results are comparable but not superior to CNN-based models in BatLiNet, despite being substantially more complex and harder to train. The added computational cost and architectural sophistication need clearer justification.

**Questions:**

- Given the higher complexity and training cost compared to CNN-based models, yet similar predictive accuracy, what concrete benefits (uncertainty calibration, generalization, or interpretability) justify using FlowBatt?

- Flow matching is claimed to be faster than diffusion; can you report actual training/sampling time or memory comparisons?

- Will you release code and pretrained models? Please ensure the repository is available at submission for reproducibility verification.

---

### Official Review · Reviewer_Pzix · 2025-11-01

**Soundness:** 2
**Presentation:** 2
**Contribution:** 2
**Rating:** 2
**Confidence:** 3

**Summary:**

The paper introduces FlowBatt, a flow-matching generative model with a Diffusion Transformer backbone that, conditioned on early-life capacity matrices, can predict state of health (SOH) trajectories and remaining useful lifetime (RUL) in batteries. On different benchmark datasets, FlowBatt delivers competitive RUL compared to other established methods and achieves the lowest SOH errors versus a supervised Transformer and a diffusion-trained variant, while quantifying uncertainty via multiple generated trajectories. The authors position FlowBatt as a scalable foundation model for battery degradation.

**Strengths:**

- The authors show that a flow-matching generative model is competitive for remaining useful life (RUL) and state of health (SOH) prediction.
- The evaluation on SOH is thorough.
- The direct, controlled comparison of training paradigms (flow matching vs diffusion vs supervised) for both SOH and RUL is interesting.
- Authors state they will release code and pretrained models, supporting reproducibility.

**Weaknesses:**

- This work might be too specialized to be appreciated by the ICLR community. I would rather see it in a specialized journal.
- The foundational model claim from the authors (Section 5) is unsupported by the experiments conducted. Why is this model positioned to be used as a foundation model compared to other solutions?
- Novelty is not clear, as there have been other works using diffusion approaches. I would advise the authors to be more clear.- For the SOH experiments, the only comparison is done with the two other training paradigms (supervised, diffusion). Why is there no comparison with other baselines?
- No intuition or explanation is given for why the performance of the flow-based network is better than the diffusion or supervised models.

**Questions:**

Feel free to address the weaknesses

---

### Official Review · Reviewer_v6bP · 2025-11-01

**Soundness:** 3
**Presentation:** 2
**Contribution:** 2
**Rating:** 2
**Confidence:** 5

**Summary:**

The paper proposes FlowBatt, a generative model for battery degradation that combines a Diffusion Transformer (DiT) backbone with a Flow Matching training objective. The model is conditioned on early-cycle capacity matrices to predict and synthesize full State-of-Health (SOH) degradation trajectories. The paper compares this approach (FlowBatt) to two internally-developed baselines: a diffusion-trained model (DiffBatt) and a supervised Transformer.

While the application of these modern techniques to battery science is interesting, the paper suffers from fatal flaws in its experimental validation and makes significant claims that are not supported by its own results. The central claim of being a "generative model" is left entirely unvalidated, and the performance claims are either unsubstantiated against external baselines or demonstrably *worse* than the current state-of-the-art.

**Strengths:**

1. The comparison of supervised, diffusion, and flow-matching objectives on the *same* DiT backbone (Tables 1 & 2) is a well-designed internal experiment. It provides good evidence that flow-matching is a superior objective *for this specific task and architecture*.
2.  The *application* of Flow Matching and DiTs to battery degradation is, to my knowledge, novel.
3.  The work in Appendix A.2 to identify and fix the instability of the CNN encoder from prior work is a good piece of engineering and diagnostic rigor.

**Weaknesses:**

### 1 The Central "Generative" Claim is Entirely Unvalidated

The paper's title and abstract frame it as a "GENERATIVE MODELING" contribution. A key stated advantage is the model's ability to "synthesize realistic degradation curves for data augmentation."

This is a central, load-bearing claim that is never validated.

* There are **no examples** of synthesized (i.e., "fake" or "novel") degradation curves anywhere in the paper or appendix. All figures (e.g., Fig 8-10) show *predictions* on the test set (i.e., `p(y|x)`), not *generations* from the learned data distribution (`p(y)`).
* There is **no quantitative evaluation** of generative quality. Standard generative model validation (e.g., using a held-out classifier to measure the realism of synthesized samples, or distribution-level metrics like MMD) is completely absent.

The paper is, in effect, a *probabilistic regression* model, not a *validated generative* model. The title and contribution claims are therefore deeply misleading.

### 2 SOH Prediction Lacks Any External Baselines

The SOH prediction results in **Table 2** are scientifically unsubstantiated for claiming SOTA or even competitive performance.

* The only models compared are `Transformer (ours)`, `DiffBatt (ours)`, and `FlowBatt (ours)`.
* The paper provides **zero** comparisons to any external, published SOTA models for SOH trajectory prediction (e.g., BatLiNet, or others from the literature).
* The conclusion that "FlowBatt further advances accuracy and generalizability, consistently outperforming both" is true *only* in this narrow, internal-only comparison. The paper proves that Flow Matching is a better training objective *for their specific architecture* than diffusion or supervised learning, but it fails to prove that FlowBatt is a good SOH predictor in the context of the wider research field. This is a critical omission.

### 3. RUL Performance is Not State-of-the-Art

The RUL prediction results in **Table 1** are presented as "comparable to or better than state-of-the-art," but this is a misrepresentation of the data.

* The current SOTA, **BatLiNet** (Zhang et al., 2025), **outperforms FlowBatt on 4 out of 5 datasets** (MATR-1, MATR-2, MIX-100, and MIX-20).
* On MATR-1, BatLiNet's RMSE is **59±2** vs. FlowBatt's **67±2**. This is a ~14% error increase.
* On MIX-100, BatLiNet's RMSE is **158±7** vs. FlowBatt's **179±6**. This is a ~13% error increase.
* On MIX-20, BatLiNet's RMSE is **201±18** vs. FlowBatt's **229±12**. This is a ~14% error increase.

The proposed model is demonstrably *less accurate* than the existing SOTA. The introduction of a highly complex generative framework (Flow Matching + DiT) has resulted in a *worse* RUL predictor. The claim of "comparable" performance is misleading.

### 4. Overstatement

Section 5, "Towards a Foundation Model," is highly speculative and misuses the term. Foundation models (Bommasani et al., 2022) are characterized by massive, broad pre-training and exceptional few-shot/zero-shot adaptation. FlowBatt is a specialized model trained on a curated set of battery datasets for a single task (degradation modeling). It shows no evidence of zero-shot adaptation or broad, emergent capabilities. This section overhypes the contribution and demonstrates a lack of rigor in the authors' claims.

**Questions:**

1.  Given the title "generative modeling", can the authors please provide (1) qualitative examples of *unconditionally synthesized* degradation curves and (2) a *quantitative* evaluation of their realism/quality?
2.  For Table 2 (SOH Prediction), can the authors please provide a comparison against at least one external, state-of-the-art baseline (e.g., BatLiNet, evaluated using the same full-trajectory RMSE metric)?
3.  How do the authors justify the claim of "comparable" RUL performance (Table 1) when their model is outperformed by BatLiNet on 4/5 datasets, often by a significant (13-14%) margin?
4.  Given the model's specialized training and task, can the authors please provide a more rigorous justification for using the term "Foundation Model," or otherwise retract this claim?

---

### Meta-Review · Area_Chair_Tmph · 2025-12-27

**Summary:**

This paper introduces FlowBatt, a scalable generative model for battery degradation prediction and synthesis that combines flow matching with a diffusion transformer (DiT) backbone. FlowBatt operates as both a probabilistic predictor of full state-of-health (SOH) trajectories and a generative tool for synthesizing realistic degradation curves, aiming to address the challenge of accurate battery cycle life prediction amid complex aging and cycling behaviors.

Reviewers’ core concerns informing the decision included: (1) the novelty and comparative advantage of flow matching over existing diffusion processes and supervised learning methods in battery degradation modeling; (2) the rigor of uncertainty quantification; (3) the generalizability of FlowBatt to diverse battery chemistries, operating conditions, and datasets beyond the evaluated benchmarks; and (4) the practical utility of generated degradation curves for data augmentation. No rebuttal is provided during the discussion period.

Overall, I recommend rejection.

**Reviewer Concerns:**

The authors consistently express concerns about the paper's novelty, insufficient evaluation, and lack of generalizability, but there is no rebuttal from the authors to address these concerns.

**Reviewer Scores:**

The reviewers all have negative ratings for this paper, and no rebuttal is provided.

---

### Decision · Program_Chairs · 2026-01-26

Reject